# ATTACKING LIFELONG LEARNING MODELS WITH GRADIENT REVERSION

## ABSTRACT

Lifelong learning aims at avoiding the catastrophic forgetting problem of traditional supervised learning models. Episodic memory based lifelong learning methods such as A-GEM (Chaudhry et al., 2018b) are shown to achieve the state-of-the-art results across the benchmarks. In A-GEM, a small episodic memory is utilized to store a random subset of the examples from previous tasks. While the model is trained on a new task, a reference gradient is computed on the episodic memory to guide the direction of the current update. While A-GEM has strong continual learning ability, it is not clear that if it can retain the performance in the presence of adversarial attacks. In this paper, we examine the robustness of A-GEM against adversarial attacks to the examples in the episodic memory. We evaluate the effectiveness of traditional attack methods such as FGSM and PGD. The results show that A-GEM still possesses strong continual learning ability in the presence of adversarial examples in the memory and simple defense techniques such as label smoothing can further alleviate the adversarial effects. We presume that traditional attack methods are specially designed for standard supervised learning models rather than lifelong learning models. we therefore propose a principled way for attacking A-GEM called *gradient reversion* (GREV) which is shown to be more effective. Our results indicate that future lifelong learning research should bear adversarial attacks in mind to develop more robust lifelong learning algorithms.

## 1 INTRODUCTION

Lifelong learning (French, 1999; Thrun & Mitchell, 1995; Kirkpatrick et al., 2017) aims at improving the continual learning ability of neural networks. Standard supervised learning methods suffer from the problem of *catastrophic forgetting*, in which case the models gradually forget previously learned knowledge while learning on a sequence of new tasks. In lifelong learning, neural networks are equipped with the capability to learn new tasks while maintaining the performance on the tasks trained previously. Lifelong learning models with continual learning ability can be deployed in complex environments with the aim to process a continuous stream of information.

Several methodologies are proposed recently to address the catastrophic forgetting problem. In Kirkpatrick et al. (2017), the authors adopt Fisher information matrix to prevent important weights for old tasks from drastic changes while the model is trained on a new task. While in Rusu et al. (2016), a neural network that has lateral connections with old tasks is trained each time for the new task. Recently, lifelong learning methods based on episodic memory (Lopez-Paz et al., 2017; Chaudhry et al., 2018b; d'Autume et al., 2019) such as A-GEM (Chaudhry et al., 2018b) are shown to be able to achieve the state-of-the-art performance across several benchmarks. In A-GEM, a small episodic memory is utilized to store a random subset of the examples from old tasks. While the model is trained on a new task, a reference gradient is computed on a batch of the samples from episodic memory to guide the current update direction. If the angle between the reference gradient and the current gradient computed on the new task is obtuse, the current gradient is projected to be perpendicular to the reference gradient.

The strong continual learning ability of A-GEM relies on the episodic memory which can give a hint on the performance of the current model on old tasks. It has been known that in the standard super-

vised learning setting, deep neural networks can be easily fooled by adversarial examples (Szegedy et al., 2013; Goodfellow et al., 2014). A natural question then arises:

> *Can* A-GEM *retain the continual learning ability in the presence of adversarial examples in the episodic memory?*

In this paper, we systematically evaluate the robustness of A-GEM against traditional adversarial attack methods such as FGSM (Goodfellow et al., 2014) and PGD (Madry et al., 2017). The results show that A-GEM is surprisingly robust under traditional adversarial attacks. We therefore propose *gradient reversion* (GREV) attack, which is a principled way for attacking episodic memory based lifelong learning algorithms such as A-GEM. Essentially, GREV alters the direction of reference gradient computed on the episodic memory by slightly perturbing the examples. Our results show that for future research on lifelong learning, it is important to design algorithms bearing adversarial attacks in mind.

In this paper, we have the following contributions,

- To the best of our knowledge, we are the first to systematically evaluate the robustness of episodic memory based lifelong learning algorithms such as A-GEM.
- We show that simple adversarial attack methods such as fast gradient sign method (FGSM) (Goodfellow et al., 2014) and projected gradient descent (PGD) (Madry et al., 2017) can hardly hurt the performance of A-GEM. Defense techniques such as label smoothing can be used to further alleviate the adversarial effect.
- We propose a principled way called gradient reversion (GREV) for attacking A-GEM. On **Permuted MNIST**, we show that A-GEM achieves an accuracy which is 40% lower than the original accuracy under the proposed GREV attack. On **Split CIFAR**, while FGSM and PGD cannot hurt the performance of A-GEM, the proposed GREV degrades the accuracy of A-GEM by as much as 20%.

## 2 BACKGROUND

### 2.1 LIFELONG LEARNING

In a lifelong learning task, suppose there are a sequence of $N$ datasets denoted as $\{D_1, D_2, ..., D_N\}$. Each dataset $D_i$ is a collection of pairs $\{\mathbf{x}_j^i, y_j^i\}$, where $\mathbf{x}_j^i$ is the $j$-th example of task $i$ and $y_j^i$ is the corresponding label. A model $f(\mathbf{w}; \mathbf{x})$ with weight $\mathbf{w}$ is trained continually on the tasks with a single pass over the examples of each task. We denote the weight as $\mathbf{w}_i$ while the model is trained on the $i$-th task and the training loss on the $i$-th task is denoted as $\ell(\mathbf{w}_i; D_i)$.

The most commonly used metric for evaluating the performance of lifelong learning models is Average Accuracy (AA), which is the average test accuracy on the test set of each task after the model finishes training on all tasks. In order to achieve a high Average Accuracy, the model should maintain the performance on old tasks while training on a new task.

### 2.2 A-GEM

In this section, we review the Averaged Gradient Episode Memory (A-GEM) (Chaudhry et al., 2018b), one of the state-of-the-art lifelong learning methods. In A-GEM, a small episodic memory $M$ with fixed size is used to store a subset of the examples from old tasks. The episodic memory is populated by choosing examples uniformly at random for each task. $M_k$ is used to denote the examples in the episodic memory from task $k$. While training on task $i$, the loss on the episodic memory $M$ can be computed as $\ell(\mathbf{w}_i; M)$, where $M = \cup_{k<i} M_k$. A-GEM ensures that each update on the $i$-th task will not increase the loss on the episodic memory, that is,

$$\min_{\mathbf{w}} \ell(\mathbf{w}; D_i) \quad \text{s.t.} \quad \ell(\mathbf{w}; M) \leq \ell(\mathbf{w}_{i-1}; M) \quad \text{where } M = \cup_{k<i} M_k \qquad (1)$$

To inspect the increase of loss on the episodic memory, A-GEM computes the gradient $\mathbf{g}$ on the current task and the reference gradient $\mathbf{g}_{\text{ref}}$ on the episodic memory. When the angle between $\mathbf{g}$ and

$\mathbf{g}_{\text{ref}}$ is obtuse, A-GEM projects the current gradient $\mathbf{g}$ to have a right or acute angle with $\mathbf{g}_{\text{ref}}$,

$$\min_{\mathbf{g}_{\text{true}}} \frac{1}{2} \|\mathbf{g} - \mathbf{g}_{\text{true}}\|_2^2 \quad \text{s.t.} \quad \mathbf{g}_{\text{true}}^\top \mathbf{g}_{\text{ref}} \geq 0 \tag{2}$$

The above optimization problem can be solved in closed form as,

$$\mathbf{g}_{\text{true}} = \mathbf{g} - \frac{\mathbf{g}^\top \mathbf{g}_{\text{ref}}}{\mathbf{g}_{\text{ref}}^\top \mathbf{g}_{\text{ref}}} \mathbf{g}_{\text{ref}} \tag{3}$$

The current gradient $\mathbf{g}$ is then replaced by $\mathbf{g}_{\text{true}}$ for updating the model.

Essentially, A-GEM avoids the catastrophic forgetting problem by altering the direction of the current gradient. When the current gradient is detrimental to old tasks, it is adjusted to be perpendicular or acute with the gradient computed on the episodic memory.

## 3 A THREAT MODEL FOR ATTACKING AND DEFENSE OF MEMORY BASED LIFELONG LEARNING ALGORITHMS

In this section, we specify the threat model used in the paper for attacking A-GEM. Specially, we consider a white-box adversary which has access to the model architecture and parameters. In addition, the adversary is allowed to perturb the examples in the episodic memory. However, we do not expose the training process to the adversary, that is, the adversary can only perturb the examples in the episodic memory in an offline fashion. Specially, before the model is trained on the $i$-th task, the adversary can slightly perturb the examples from task $i-1$ in the episodic memory. In this setting, each example is only perturbed once during the lifelong learning process. We refer to this threat model as Offline Sequential Attack. The defender, on the other hand, has access to the reference gradients but not the perturbed examples in the episodic memory. The defender is further allowed to take advantage of the labels of the perturbed examples in the memory for defending possible attacks. The proposed threat model is a generalized version of white-box adversaries (Goodfellow et al., 2014; Carlini & Wagner, 2017a) for lifelong learning setting. The Average Accuracy of the lifelong learning model with and without adversarial attack is referred to as perturbed accuracy and unperturbed accuracy respectively.

## 4 TRADITIONAL ATTACK METHODS

The objective of traditional adversaries is to find an adversarial example $\mathbf{x}_{\text{adv}}$ for $\mathbf{x}$ such that they are imperceptibly close and yet the neural network labels them distinctly. We bound the $\ell_p$ distance between an input $\mathbf{x}$ and its adversarial counterpart: $\mathbf{x}_{\text{adv}} \in S_p(\mathbf{x}) := \{\mathbf{x}' : \|\mathbf{x} - \mathbf{x}'\|_p \leq \tau_p\}$, where $p = 2$ or $\infty$. We omit from $S_p(\mathbf{x})$ the argument $\mathbf{x}$ and the subscript $p$ when it does not cause ambiguity. We focus on $\ell_\infty$ bounded attack in this paper since $\ell_\infty$ distance has been shown as a natural metric to measure adversarial perturbations.

**FGSM** Fast Gradient Sign Method (FGSM) (Goodfellow et al., 2014) is a simple one-step adversarial attack method. It is featured by efficiency and high performance for generating $\ell_\infty$ bounded adversarial examples. In FGSM, the adversarial counterpart of example $\mathbf{x}$ is produced by,

$$\mathbf{x} + \epsilon \text{sign}(\nabla_{\mathbf{x}} \ell(\mathbf{w}; \mathbf{x})) \tag{4}$$

**PGD** A more powerful attack technique has been proposed as a multi-step variant of FGSM, which is called projected gradient descent (PGD) (Madry et al., 2017),

$$\mathbf{x}_{n+1} = \Pi_S(\mathbf{x}_n + \eta \text{sign}(\nabla_{\mathbf{x}_n} \ell(\mathbf{w}; \mathbf{x}_n))) \tag{5}$$

where $S = \{\mathbf{x} : \|\mathbf{x} - \mathbf{x}_n\| \leq \tau\}$ and $\Pi$ is the projection operator.

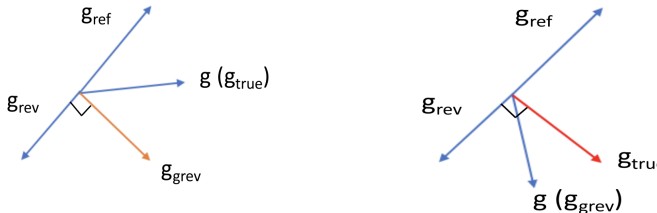

(a) Case1: the angle between the current gradient **g** and the reference gradient **g**$_{\text{ref}}$ is acute.

(b) Case2: the angle between the current gradient **g** and the reference gradient **g**$_{\text{ref}}$ is obtuse.

Figure 1: The illustration of our proposed gradient reversion attack.

**Rotation** We also consider a simple variant of the rotation attack introduced in Engstrom et al. (2017). If the example **x** is rotated by an angle $\theta$, the pixel at position $(u, v)$ is translated to $(u', v')$ via the following equation,

$$\begin{bmatrix} u' \\ v' \end{bmatrix} = \begin{bmatrix} \cos\theta & -\sin\theta \\ \sin\theta & \cos\theta \end{bmatrix} \cdot \begin{bmatrix} u \\ v \end{bmatrix} \tag{6}$$

## 5 A PRINCIPLED WAY OF ATTACKING MEMORY BASED LIFELONG LEARNING METHODS

The traditional attack methods, such as FGSM and PGD, are designed based on the idea that the goal of attacking is to degrade the accuracy of the model by appropriately perturbing the examples. While in lifelong learning, the aim is to perturb the examples in the memory in the way that the model cannot retain performance on old tasks. Dominated by different motivations, traditional attack methods may not be suitable for attacking lifelong learning models.

Note that in the A-GEM framework, as illustrated in Section 2.2, the angle between the stochastic gradient of the current task and the episodic memory is computed in each iteration. If the angle is obtuse, then A-GEM adjusts the current update direction appropriately. The main idea of our attack mechanism is to disseminate misinformation by providing a corrupted reference gradient, which can be realized by manipulating the examples stored in the episodic memory in a specific way. Based on this idea, we propose an attack method which tries to find an appropriate perturbation to the examples in the episodic memory to minimize the correlation (characterized by inner product) between the vanilla reference gradient and the corrupted reference gradient calculated on the perturbed data. Mathematically, the objective function can be written as,

$$\min_{\|\delta\| \leq \tau} \langle \nabla_{\mathbf{w}} \ell(\mathbf{w}; \mathbf{x}), \nabla_{\mathbf{w}} \ell(\mathbf{w}; \mathbf{x} + \delta) \rangle, \tag{7}$$

where **w** is the trainable parameter of the model, **x** is an example in the episodic memory, $\ell$ is the loss function, $\tau > 0$ is a hyper-parameter which characterizes the feasible region of the perturbation. If $\tau$ is large enough, there exists at least one sub-optimal perturbation $\tilde{\delta}$ such that $\nabla_{\mathbf{w}} \ell(\mathbf{w}; \mathbf{x} + \tilde{\delta})$ is in the opposite direction of $\nabla_{\mathbf{w}} \ell(\mathbf{w}; \mathbf{x})$. In Figure 1, **g**, **g**$_{\text{ref}}$, **g**$_{\text{rev}}$, **g**$_{\text{true}}$, **g**$_{\text{grev}}$ stand for stochastic gradient calculated on the current task, stochastic gradient calculated on the examples in episodic memory (i.e., the reference gradient $\nabla_{\mathbf{w}} \ell(\mathbf{w}; \mathbf{x})$), the corrupted reference gradient (i.e., $\nabla_{\mathbf{w}} \ell(\mathbf{w}; \mathbf{x} + \tilde{\delta})$), the true update direction found by A-GEM on the unperturbed memory, the update direction found by A-GEM under gradient reversion attack. Below we analyze two common cases during the learning process,

- In case 1, the angle between the current gradient **g** and the reference gradient **g**$_{\text{ref}}$ is acute. The true update direction **g**$_{\text{true}}$ found by A-GEM is coincident with **g**. With the proposed gradient reversion attack, the actual update direction **g**$_{\text{grev}}$ is perpendicular to **g**$_{\text{ref}}$. Although the update direction **g**$_{\text{grev}}$ will not lead to an increase of loss on the episodic memory, it does not help the learning on old tasks as **g**$_{\text{true}}$ does.

- In case 2, the angle between the current gradient $\mathbf{g}$ and the reference gradient $\mathbf{g}_{\text{ref}}$ is obtuse. The true update direction $\mathbf{g}_{\text{true}}$ found by A-GEM is perpendicular to $\mathbf{g}_{\text{ref}}$. Since $\mathbf{g}_{\text{grev}}$ is coincident with $\mathbf{g}$ in this case, the update direction $\mathbf{g}_{\text{grev}}$ will deteriorate the performance of the model on the episodic memory.

In both cases, we can see that $\mathbf{g}_{\text{grev}}$ is either perpendicular or negatively correlated to $\mathbf{g}_{\text{ref}}$, which directly leads to catastrophic forgetting.

To solve (7), we can utilize projected gradient descent update for multiple iterations:

$$\delta \leftarrow \Pi_{\|\delta\| \leq \tau} \left[ \delta - \eta \left( \left( \nabla_\delta \nabla_{\mathbf{w}} \ell(\mathbf{w}; \mathbf{x} + \delta) \right)^\top \nabla_{\mathbf{w}} \ell(\mathbf{w}; \mathbf{x}) \right) \right] \tag{8}$$

It is worth mentioning that the update (8) can be implemented efficiently. It requires three back-propagations in the first iteration, and only two back-propagations thereafter. Specifically, in the first iteration, we need to evaluate $\nabla_{\mathbf{w}} \ell(\mathbf{w}; \mathbf{x})$ and $\nabla_{\mathbf{w}} \ell(\mathbf{w}; \mathbf{x} + \delta)$ with two back-propagations, and one more back-propagation is needed to calculate the gradient with respect to $\delta$. In the subsequent iterations, $\nabla_{\mathbf{w}} \ell(\mathbf{w}; \mathbf{x})$ is already available and only two back-propagations are needed. Note that our computational cost per iteration is almost the same as PGD. The reason is that the dimension of $\delta$ is much smaller than that of $\mathbf{w}$, so the additional computational overhead is negligible.

# 6 EXPERIMENTAL SETTINGS

In the experiments, we are interested in investigating the following questions,

- Can traditional attack methods successfully attack A-GEM?
- How effective is our proposed attack method compared with traditional attack methods?
- How is the perturbed accuracy affected by the size of the episodic memory?

## 6.1 DATASETS

We use **Permuted MNIST** and **Split CIFAR** in the experiments. **Permuted MNIST** (Kirkpatrick et al., 2017) consists of 20 tasks and each task is constructed by applying the same permutation to the pixel of examples in the MNIST dataset (LeCun et al., 1998). **Split CIFAR** (Kirkpatrick et al., 2017) is constructed by splitting the original CIFAR100 dataset (Krizhevsky et al., 2009) into 20 disjoint sets. Each set is constructed by randomly sampling 5 categories of the dataset.

## 6.2 NETWORK ARCHITECTURES

We use the same network architectures as in Chaudhry et al. (2018b). For **Permuted MNIST**, we adopt a fully-connected neural network with two hidden layers of 256 ReLU units. For **Split CIFAR**, we use a reduced ResNet18 (He et al., 2016).

## 6.3 EVALUATION PROTOCOL

We follow the same training settings as in Chaudhry et al. (2018b). For **Permuted MNIST**, the maximum perturbation $\tau$ is selected from $\{0.05, 0.1, 0.2\}$. The episodic memory size is set to be $\{850, 1700, 2550, 4250\}$, which corresponds to 5, 10, 15, 25 examples per class. For **Split CIFAR**, the maximum perturbation $\tau$ is selected from $\{0.015, 0.031, 0.055\}$. The episodic memory size is set to be $\{425, 850, 1105\}$, which corresponds to 5, 10, 13 examples per class. Both PGD and GREV are iterated for 40 steps. In the rotation attack, the examples in the episode memory are rotated by 3, 5, 7 degrees. All the experiments are repeated for 5 times with different random seeds and the variance is reported.

# 7 RESULTS

## 7.1 PERMUTED MNIST

We show the results of the perturbed accuracy of different attack methods and the unperturbed accuracy without attack on **Permuted MNIST** in Table 1. The results show several intriguing properties

| | Memory Size | $\tau = 0.05$ | $\tau = 0.1$ | $\tau = 0.2$ | No Attack |
|---|---|---|---|---|---|
| FGSM | 4250 | $0.872 \pm 0.002$ | $0.785 \pm 0.010$ | $0.510 \pm 0.02$ | $0.892 \pm 0.003$ |
| | 2550 | $0.866 \pm 0.007$ | $0.788 \pm 0.004$ | $0.543 \pm 0.006$ | $0.885 \pm 0.004$ |
| | 1700 | $0.858 \pm 0.002$ | $0.787 \pm 0.005$ | $0.555 \pm 0.010$ | $0.875 \pm 0.003$ |
| | 850 | $0.831 \pm 0.004$ | $0.761 \pm 0.008$ | $0.550 \pm 0.016$ | $0.861 \pm 0.001$ |
| | Memory Size | $\tau = 0.05$ | $\tau = 0.1$ | $\tau = 0.2$ | No Attack |
| PGD | 4250 | $0.868 \pm 0.004$ | $0.765 \pm 0.006$ | $0.471 \pm 0.013$ | $0.892 \pm 0.003$ |
| | 2550 | $0.863 \pm 0.004$ | $0.771 \pm 0.012$ | $0.511 \pm 0.011$ | $0.885 \pm 0.004$ |
| | 1700 | $0.852 \pm 0.003$ | $0.768 \pm 0.007$ | $0.523 \pm 0.009$ | $0.875 \pm 0.003$ |
| | 850 | $0.829 \pm 0.003$ | $0.749 \pm 0.007$ | $0.543 \pm 0.004$ | $0.861 \pm 0.001$ |
| | Memory Size | $deg = 3$ | $deg = 5$ | $deg = 7$ | No Attack |
| Rotation | 4250 | $0.805 \pm 0.009$ | $0.659 \pm 0.018$ | $0.597 \pm 0.010$ | $0.892 \pm 0.003$ |
| | 2550 | $0.804 \pm 0.012$ | $0.651 \pm 0.019$ | $0.591 \pm 0.022$ | $0.885 \pm 0.004$ |
| | 1700 | $0.789 \pm 0.008$ | $0.644 \pm 0.020$ | $0.589 \pm 0.011$ | $0.875 \pm 0.003$ |
| | 850 | $0.748 \pm 0.012$ | $0.602 \pm 0.012$ | $0.550 \pm 0.019$ | $0.861 \pm 0.001$ |
| | Memory Size | $\tau = 0.05$ | $\tau = 0.1$ | $\tau = 0.2$ | No Attack |
| GREV | 4250 | $0.421 \pm 0.016$ | $0.450 \pm 0.019$ | $0.474 \pm 0.018$ | $0.892 \pm 0.003$ |
| | 2550 | $0.440 \pm 0.015$ | $0.482 \pm 0.011$ | $0.508 \pm 0.017$ | $0.885 \pm 0.004$ |
| | 1700 | $0.467 \pm 0.019$ | $0.469 \pm 0.022$ | $0.488 \pm 0.014$ | $0.875 \pm 0.003$ |
| | 850 | $0.451 \pm 0.012$ | $0.472 \pm 0.017$ | $0.494 \pm 0.005$ | $0.861 \pm 0.001$ |

Table 1: The perturbed accuracy of different attack methods on **Permuted MNIST**.

of attacking lifelong learning models compared with attacking standard supervised learning models. When $\tau = 0.05$, both FGSM and PGD can hardly hurt the performance of A-GEM which shows the surprising robustness of A-GEM. Although the model performs much worse on the episodic memory after the attack, we presume that the direction of the gradient on the episodic memory remains nearly unchanged which allows A-GEM to derive the correct update direction. Only when the value of $\tau$ increases to 0.2, FGSM and PGD can achieve a much lower perturbed accuracy. For the rotation attack, rotating the examples in the memory by only 3 degrees results in a large drop in accuracy. Intuitively, by rotating the examples, the direction of the gradient changes accordingly which fools A-GEM to conduct incorrect projections. Compared with the traditional attack methods, the proposed gradient reversion attack (GREV) achieves a much lower perturbed accuracy even when $\tau$ equals to 0.05. This indicates that by directly perturbing the examples in the way to reverse the direction of reference gradient, we can achieve a much more effective attack.

## 7.2 SPLIT CIFAR

We show the results of perturbed accuracy of different attack methods and the unperturbed accuracy without attack on **Split CIFAR** in Table 2. The experiments on **Split CIFAR** allow us to examine how different attack methods behave with convolutional neural networks. Surprisingly, we observe that FGSM and PGD cannot attack A-GEM in this case even with a large $\tau$. And the rotation attack, which is effective on **Permuted MNIST**, cannot attack A-GEM with convolutional neural networks either. On the other hand, the proposed gradient reversion attack successfully degrades the accuracy of A-GEM by about 20%. We show in Section 7.3 that PGD can hardly alter the direction of the reference gradient which explains the ineffectiveness of PGD in this case. While the proposed GREV attack drastically change the direction of the reference gradient which leads to a large drop in accuracy. Under the proposed GREV attack, we observe that the perturbed accuracy is roughly inversely proportional to the size of the episodic memory. This indicates that although A-GEM can achieve higher unperturbed accuracy with large episodic memory, it also suffers more in the presence of adversarial examples.

## 7.3 ANALYSIS OF PGD AND GREV

We now compare the angle between the reference gradient $\mathbf{g}_{\text{ref}}$ on the unperturbed data and the corrupted reference gradient on the perturbed data under PGD and GREV attack. This allows us to gain more insights on how different attack methods behave. On **Permuted MNIST**, the size of the episodic memory is set to be 4250 and $\tau = 0.05$. On **Split CIAFR**, the size of the episodic memory is set to be 1105 and $\tau = 0.015$. In Figure 2, we show the distribution of the angles between the

| FGSM | Memory Size | $\tau = 0.015$ | $\tau = 0.031$ | $\tau = 0.055$ | No Attack |
|---|---|---|---|---|---|
| | 1105 | $0.601 \pm 0.025$ | $0.587 \pm 0.021$ | $0.587 \pm 0.020$ | $0.585 \pm 0.016$ |
| | 850 | $0.595 \pm 0.017$ | $0.589 \pm 0.033$ | $0.589 \pm 0.034$ | $0.593 \pm 0.027$ |
| | 425 | $0.591 \pm 0.024$ | $0.591 \pm 0.024$ | $0.559 \pm 0.018$ | $0.590 \pm 0.025$ |
| PGD | Memory Size | $\tau = 0.015$ | $\tau = 0.031$ | $\tau = 0.055$ | No Attack |
| | 1105 | $0.587 \pm 0.019$ | $0.571 \pm 0.017$ | $0.571 \pm 0.017$ | $0.585 \pm 0.016$ |
| | 850 | $0.572 \pm 0.030$ | $0.587 \pm 0.018$ | $0.587 \pm 0.017$ | $0.593 \pm 0.027$ |
| | 425 | $0.588 \pm 0.021$ | $0.588 \pm 0.021$ | $0.588 \pm 0.020$ | $0.590 \pm 0.025$ |
| Rotation | Memory Size | $deg = 3$ | $deg = 5$ | $deg = 7$ | No Attack |
| | 1105 | $0.584 \pm 0.028$ | $0.586 \pm 0.019$ | $0.592 \pm 0.012$ | $0.585 \pm 0.016$ |
| | 850 | $0.570 \pm 0.024$ | $0.594 \pm 0.032$ | $0.578 \pm 0.018$ | $0.593 \pm 0.027$ |
| | 425 | $0.586 \pm 0.020$ | $0.571 \pm 0.039$ | $0.575 \pm 0.022$ | $0.590 \pm 0.025$ |
| GREV | Memory Size | $\tau = 0.015$ | $\tau = 0.031$ | $\tau = 0.055$ | No Attack |
| | 1105 | $0.382 \pm 0.046$ | $0.381 \pm 0.045$ | $0.397 \pm 0.027$ | $0.585 \pm 0.016$ |
| | 850 | $0.393 \pm 0.037$ | $0.406 \pm 0.039$ | $0.406 \pm 0.040$ | $0.593 \pm 0.027$ |
| | 425 | $0.413 \pm 0.031$ | $0.415 \pm 0.035$ | $0.414 \pm 0.035$ | $0.590 \pm 0.025$ |

Table 2: The perturbed accuracy of different attack methods on **Split CIFAR**.

reference gradient on 200 random batches of unperturbed examples from task 1 and the corrupted reference gradient on the corresponding perturbed examples under PGD and GREV attack during the training of task 2. We can see that the proposed GREV attack is a more effective way to alter the direction of the reference gradient, especially on **Split CIAFR**. In Figure 3, we can see that the angle between the reference gradient and the corrupted reference gradient gradually increases under the GREV attack. The results show that the proposed GREV attack enjoys some special properties which make it a more effective way for attacking A-GEM.

## 8 RELATED WORK

### 8.1 LIFELONG LEARNING

Recent lifelong learning works mostly focus on regularization based lifelong learning methods (Kirkpatrick et al., 2017; Zenke et al., 2017; Chaudhry et al., 2018a) and episodic memory based lifelong learning methods (Lopez-Paz et al., 2017; Chaudhry et al., 2018b; d'Autume et al., 2019). In EWC (Kirkpatrick et al., 2017), Fisher information matrix is adopted to prevent important weights for old tasks from drastic change. Zenke et al. (2017) introduced *intelligent synapse* which has a local measure of "importance" to avoid old memories from being overwritten. RWALK (Chaudhry et al., 2018a) leverages a KL-divergence for retaining knowledge for old tasks. In (Lopez-Paz et al., 2017), the authors introduced *gradient episodic memory* (GEM) which achieves the state-of-the-art results on several benchmarks. In (Chaudhry et al., 2018b). the authors developed an efficient version of GEM called A-GEM which is more computational effective. d'Autume et al. (2019) generalized episodic memory based methods for lifelong language modelling.

### 8.2 ADVERSARIAL ATTACK

Adversarial attack techniques can be briefly divided into two categories: white-box attack and black-box attack. Regarding to the defense aspect, we refer readers to Appendix A.2 for some recent works.

**White-box attack.** Preliminary studies on the robustness of DNNs focus on white-box setting with assuming full access to the targeted DNN. Szegedy et al. (2013) first prove DNN is fragile against adversarial examples and generate adversarial examples $\mathbf{x}'$ similar to original sample $\mathbf{x}$ in $\ell_2$ distance using box-constrained *L-BFGS*. Then the fast gradient sign (*FGS*) (Goodfellow et al., 2014) method has been invented for efficiently producing adversarial examples in $\ell_\infty$ distance. Papernot et al. (2016) introduce an attack optimized under $l_0$ distance known as the Jacobian-based Saliency Map Attack (*JSMA*). *DeepFool* (Moosavi-Dezfooli et al., 2016) is an untargeted attack algorithm that aims to find the least $\ell_2$ distortion leading to misclassification by projecting an image to the closest separating hyperplane. Following these works, Carlini & Wagner (2017b) proposed an iter-

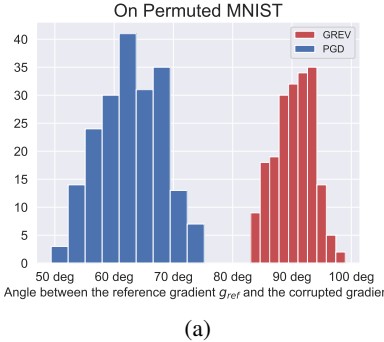 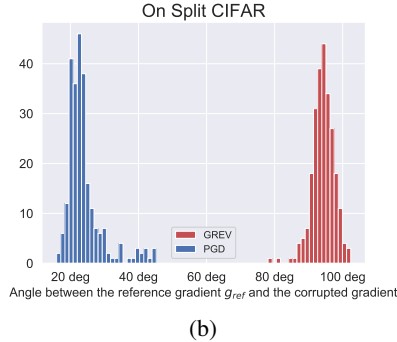

Figure 2: Comparison of GREV and PGD with respect to the ability of altering the reference gradient direction.

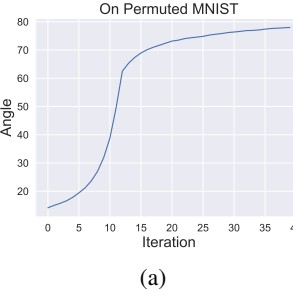 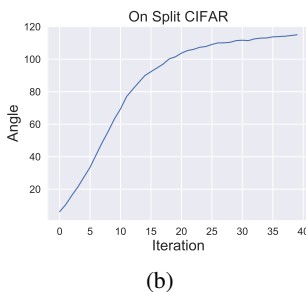

Figure 3: The angle between the reference gradient and the corrupted reference gradient gradually increases under the proposed GREV attack.

ative optimization based attack (*C&W attack*), and then it seems to become a standard white-box attack approach. Similarly, projected gradient descent (PGD) has been shown strong in attacking DNNs (Madry et al., 2017). Most of the white-box attacks rely on the gradients of the DNNs. When the gradients are "obfuscated" (e.g., by randomization), Athalye et al. (2018) derive various methods to approximate the gradients.

**Black-box attack.** The black-box attacking techniques do not exert the internal knowledge of DNN, and are more practical in the real applications. Thanks to the transferability property of adversarial examples (Szegedy et al., 2013), Papernot et al. (2017) can train a substitute DNN to imitate the behavior of the unknown DNN to be attacked, produce adversarial examples of the substitute, and then use them to attack. Chen et al. (2017) instead use zero-th order optimization to find adversarial examples. Ilyas et al. (2018) use the evolution strategy (Salimans et al., 2017) to approximate the gradients. More recently, Brendel et al. (2017) introduce Boundary Attack which starts from a large adversarial perturbation and then seeks to reduce the perturbation while staying adversarial. Li et al. (2019) proposed a universal attack for defended DNNs by modelling the distributions of adversarial examples.

## 9 CONCLUSION

In this paper, we systematically examine the robustness of episodic memory lifelong learning methods such as A-GEM under traditional adversarial attacks. The results show that different from traditional supervised learning model, A-GEM is surprisingly robust to these attacks. We therefore propose an attack named Gradient Reversion (GREV) which makes A-GEM suffer from significant performance degradation. In the future, we plan to design defense mechanisms to mitigate the negative effects caused by the gradient reversion attack and develop more robust lifelong learning algorithms.

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

| | Memory Size | $\tau = 0.05$ | $\tau = 0.1$ | $\tau = 0.2$ | No Attack |
|---|---|---|---|---|---|
| PGD | 4250 | $0.853 \pm 0.004$ | $0.823 \pm 0.004$ | $0.724 \pm 0.007$ | $0.892 \pm 0.003$ |
| | 2550 | $0.855 \pm 0.003$ | $0.825 \pm 0.003$ | $0.735 \pm 0.008$ | $0.885 \pm 0.004$ |
| | 1700 | $0.864 \pm 0.003$ | $0.814 \pm 0.008$ | $0.705 \pm 0.006$ | $0.875 \pm 0.003$ |
| | 850 | $0.869 \pm 0.002$ | $0.819 \pm 0.004$ | $0.772 \pm 0.007$ | $0.861 \pm 0.001$ |
| | Memory Size | $\tau = 0.05$ | $\tau = 0.1$ | $\tau = 0.2$ | No Attack |
| GREV | 4250 | $0.428 \pm 0.011$ | $0.445 \pm 0.015$ | $0.473 \pm 0.017$ | $0.892 \pm 0.003$ |
| | 2550 | $0.436 \pm 0.019$ | $0.463 \pm 0.025$ | $0.506 \pm 0.010$ | $0.885 \pm 0.004$ |
| | 1700 | $0.476 \pm 0.015$ | $0.471 \pm 0.026$ | $0.483 \pm 0.022$ | $0.875 \pm 0.003$ |
| | 850 | $0.437 \pm 0.022$ | $0.461 \pm 0.018$ | $0.468 \pm 0.020$ | $0.861 \pm 0.001$ |

Table 3: PGD and GREV on **Permuted MNIST** with label smoothing parameter $\beta = 0.1$.

| Memory Size | $\tau = 0.015$ | $\tau = 0.031$ | $\tau = 0.055$ | No Attack |
|---|---|---|---|---|
| 1105 (13 * 5 * 17) | $0.416 \pm 0.036$ | $0.415 \pm 0.035$ | $0.383 \pm 0.019$ | $0.585 \pm 0.016$ |
| 850 (10 * 5 * 17) | $0.414 \pm 0.032$ | $0.413 \pm 0.031$ | $0.385 \pm 0.024$ | $0.593 \pm 0.027$ |
| 425 (5 * 5 * 17) | $0.418 \pm 0.036$ | $0.417 \pm 0.036$ | $0.401 \pm 0.047$ | $0.590 \pm 0.025$ |

Table 4: GREV on **Split CIFAR** with label smoothing parameter $\beta = 0.1$.

## A    Appendix

### A.1    Defense Lifelong Learning Models

In this section, we investigate a possible defense technique called label smoothing (Warde-Farley & Goodfellow, 2016; Müller et al., 2019) for defensing the attacks to A-GEM. Label smoothing converts hard class labels into soft targets as follows,

$$y_{LS}^k = y_k(1 - \beta) + \frac{\beta}{K} \tag{9}$$

where $y_k$ is the one-hot label of class $k$, $K$ is the number of classes, $\beta$ is the label smoothing parameter and $y_{LS}^k$ is the new one-hot label after label smoothing. Label smoothing has been shown as a simple way for increasing the robustness of neural networks (Warde-Farley & Goodfellow, 2016).

Instead of using labeling smoothing to train the model, we convert the labels of the examples in the memory to soft labels via label smoothing. The results on **Permuted MNIST** are shown in Table 3. It can be seen that label smoothing can further increase the robustness of A-GEM against PGD attack. However, the proposed GREV attack can still achieve a low perturbed accuracy. Table 4 also shows that the proposed GREV attack can greatly deteriorate the performance of A-GEM on **Split CIFAR** with label smoothing. Therefore, one future research direction is to develop defense methods for the proposed GREV attack and design more robust lifelong learning algorithms.

### A.2    Related work about defense techniques

Comparing with attacking DNNs, defending neural network is a more challenging task and a few explorations have been exerted to improve the robustness of DNNs.

Since the powerful attacking approaches take advantage of the gradient of DNNs (Carlini & Wagner, 2017b; Madry et al., 2017) or estimated gradient (Chen et al., 2017; Salimans et al., 2017), several defended approaches have been shown to be robust against these kind of attacks by 'obfuscating gradient'. Specifically, Buckman et al. (2018) proposed to transform the input by non-differentiable and non-linear thermometer encoding, followed by a slight change to the input layer of conventional DNNs. Dhillon et al. (2018) randomly dropped some neurons of each layer with the probabilities in proportion to their absolute values. Xie et al. (2018) added a randomization layer between inputs and a DNN classifier. Similarly, Liu et al. (2018) combine the ideas of randomness and ensemble using the same underlying neural network. Guo et al. (2018) explored several combinations of input transformations coupled with adversarial training, such as image cropping and rescaling, bit-depth reduction, JPEG compression. Prakash et al. (2018) randomly sample a pixel from an image and then replace it with another pixel randomly sampled from the former's neighborhood.

However, Athalye et al. (2018) have derived various approaches to estimate the gradients and successfully or partially attacked such defenses caused 'obfuscated gradient', and they have honored adversarial training based defenses. Basically, defending DNNs with adversarial training was firstly introduced by (Madry et al., 2017). The training procedure alternates between seeking an "optimal" adversarial example for each input by projected gradient descent (PGD) and minimizing the classification loss under the PGD attack. Furthermore, Na et al. (2018) reduced the computation cost of the adversarial training (Kurakin et al., 2016) in a cascade manner. Liu et al. (2019) proposed to model the randomness added to DNNs in a Bayesian framework coupled with adversarial training. Wang & Yu (2019) proposed to model the adversarial perturbation with a generative network, and they learned it jointly with the defensive DNN as a discriminator. Liao et al. (2018) use a denoising network architecture to estimate the additive adversarial perturbation to an input.

