# OpenReview forum: "Attacking Lifelong Learning Models with Gradient Reversion"
_ICLR.cc/2020/Conference — Reject_

### Official Review · AnonReviewer3 · 2019-10-22
**Official Blind Review #3**

**Rating:** 3

**Review:**

This paper does a good job of raising awareness of adversarial attacks in lifelong learning research with deep neural networks. This is the first time I have considered this problem, but not sure whether any prior work exists in the specific subfield.

At the conceptual level, many issues can arise when a lifelong learner is attacked, since systematic negative bias could be introduced in the training process and may be very difficult to remove, given the tendency to 'remember everything' which dominates current approaches.

The paper isolates one lifelong learning approach (A-GEM) which is characteristic of one (of many) different approaches to lifelong learning, and investigates its robustness to standard adversarial attacks and a novel attack developed within this paper, which is stronger, but specific to episodic memory approaches.

I cannot recommend acceptance at this point for the following reasons:
1) I am not sure what I can generalize away from this paper to the immediate subfield and beyond. The paper claims that the investigated method is SOTA, but it's not clear this is the case, even in restricted class of similar episodic memory based models, see [1] for an independent evaluation of many such approaches. Is there any reasons why conclusions about this particular method are indeed representative of its class?
2) While the paper does not explicitly make this claim, the title suggests that 'gradient reversion' attacks apply to lifelong learning models in general. Why is this class of approaches particularly informative such that conclusions may hold in general? Are other methods in this class more susceptible to these attacks and can the proposed attack be applied to the whole class, or even other types of approaches? This should be clarified!


References
[1] Matthias De Lange, Rahaf Aljundi, Marc Masana, Sarah Parisot, Xu Jia, Ales Leonardis, Gregory Slabaugh, Tinne Tuytelaars,  Continual learning: A comparative study on how to defy forgetting in classification tasks, https://arxiv.org/abs/1909.08383

**Experience Assessment:**

I have published one or two papers in this area.

**Review Assessment: Checking Correctness Of Derivations And Theory:**

I assessed the sensibility of the derivations and theory.

**Review Assessment: Checking Correctness Of Experiments:**

I did not assess the experiments.

**Review Assessment: Thoroughness In Paper Reading:**

I read the paper at least twice and used my best judgement in assessing the paper.

---

### Official Review · AnonReviewer2 · 2019-10-23
**Official Blind Review #2**

**Rating:** 3

**Review:**

The paper proposed a novel approach for robust continual learning model from the adversarial attack. The authors start from one of the state-of-the-art episodic memory based continual learning method, A-GEM. The proposed method, Gradient Reversion (GREV), is specialized on A-GEM. The method perturbed the episodic memory examples on A-GEM, thus modifies the direction of reference gradient. While conventional attack like Fast Gradient Sign Method (FGSM) and Projected Gradient Descent (PGD) hardly show the their influence on A-GEM, GREV significantly attacks the performance.


The paper is well written, and easy to follow. Also, attack technique on episodic memory based continual learning is interesting and would be valuable.

But, I feel that some of the analysis are obvious which are not much meaningful to analyze the model,  and the overall contributions are suggested under A-GEM model, while not to cover generic other episodic-based continual learning.

So, I hesitate to give the high score even the approach is interesting.

Additional one question.
I didn't get the concrete reasons that A-GEM is already robust for famous attack methods. What’s the reason that A-GEM is robust for them?


**Experience Assessment:**

I have published one or two papers in this area.

**Review Assessment: Checking Correctness Of Derivations And Theory:**

I assessed the sensibility of the derivations and theory.

**Review Assessment: Checking Correctness Of Experiments:**

I assessed the sensibility of the experiments.

**Review Assessment: Thoroughness In Paper Reading:**

I read the paper at least twice and used my best judgement in assessing the paper.

---

### Official Review · AnonReviewer1 · 2019-10-27
**Official Blind Review #1**

**Rating:** 3

**Review:**

General:
The paper first proposes an adversarial attack on the exemplar-based continual learning algorithm, A-GEM. The problem formulation is new and interesting, but I am not sure how practical or realistic the setting is. The attacker assumes to have access not only to the model but also to the episodic memory. I think that is quite a powerful assumption for the attacker and it is not very surprising that the attack would work. So, I am a bit torn with the judgments.

Summary & Pro:
1. First proposal of adversarial attack of continual learning algorithm.
2. The conventional attack schemes are shown to not work well, and they devised new one (GREV) tailored for A-GEM.
3. Experimental results show convincing results that their method works well.

Con & Questions:
1. It is not clear whether the proposed method will also work well for other exemplar-based methods like iCaRL or GEM (the simpler version than A-GEM), etc. I get that A-GEM can be attacked by their assumption and method, but how general is it?
2. What exactly is the practical scenario of this method? How can the attacker get access to the model & memory? In the traditional adversarial attack literature, it is shown that white-box attack can also lead to the black-box attack. But, in this case, I am not sure about the practical implication of the proposed methods.
3. It seems like the entire memory is under attack. What happens when only the memory is partially attacked, e.g., 10% of the data in the memory is attacked?
4. Table 1/2 only shows the overall average accuracy. Can you also show the per-task average accuracy curves? It would be much better to see such curves to clearly see the effect of the attack rather than the overall average.


**Experience Assessment:**

I have published one or two papers in this area.

**Review Assessment: Checking Correctness Of Derivations And Theory:**

I assessed the sensibility of the derivations and theory.

**Review Assessment: Checking Correctness Of Experiments:**

I assessed the sensibility of the experiments.

**Review Assessment: Thoroughness In Paper Reading:**

I read the paper at least twice and used my best judgement in assessing the paper.

---

### Decision · Program_Chairs · 2019-12-19

**Decision:**

Reject

**Comment:**

The paper investigates questions around adversarial attacks in a continual learning algorithm, i.e., A-GEM. While reviewers agree that this is a novel topic of great importance, the contributions are quite narrow, since only a single model (A-GEM) is considered and it is not immediately clear whether this method transfers to other lifelong learning models (or even other models that belong to the same family as A-GEM). This is an interesting submission, but at the moment due to its very narrow scope, it seems more appropriate as a workshop submission investigating a very particular question (that of attacking A-GEM). As such, I cannot recommend acceptance.